# Multimorbidity in Difficult Asthma: The Need for Personalised and Non-Pharmacological Approaches to Address a Difficult Breathing Syndrome

**DOI:** 10.3390/jpm12091435

**Published:** 2022-08-31

**Authors:** Judit Varkonyi-Sepp, Anna Freeman, Ben Ainsworth, Latha Perunthadambil Kadalayil, Hans Michael Haitchi, Ramesh J. Kurukulaaratchy

**Affiliations:** 1School of Clinical and Experimental Sciences, Faculty of Medicine, University of Southampton, Southampton SO16 6YD, UK; 2National Institute for Health Research (NIHR) Southampton Biomedical Research Centre, University Hospital Southampton NHS Foundation Trust, Southampton SO16 6YD, UK; 3Clinical Health Psychology Department, University Hospital Southampton NHS Foundation Trust, Southampton SO16 6YD, UK; 4Respiratory Medicine Department, University Hospital Southampton NHS Foundation Trust, Southampton SO16 6YD, UK; 5Department of Psychology, University of Bath, Bath BA2 7AY, UK; 6Institute for Life Sciences, University of Southampton, Southampton SO16 6YD, UK; 7The David Hide Asthma & Allergy Research Centre, St Mary’s Hospital, Isle of Wight, Newport PO30 5TG, UK

**Keywords:** asthma, co-morbidity, multimorbidity, personalized, non-pharmacological, therapy, treatment, treatable traits, difficult breathing syndrome, holistic treatment

## Abstract

Three to ten percent of people living with asthma have difficult-to-treat asthma that remains poorly controlled despite maximum levels of guideline-based pharmacotherapy. This may result from a combination of multiple adverse health issues including aggravating comorbidities, inadequate treatment, suboptimal inhaler technique and/or poor adherence that may individually or collectively contribute to poor asthma control. Many of these are potentially “treatable traits” that can be pulmonary, extrapulmonary, behavioural or environmental factors. Whilst evidence-based guidelines lead clinicians in pharmacological treatment of pulmonary and many extrapulmonary traits, multiple comorbidities increase the burden of polypharmacy for the patient with asthma. Many of the treatable traits can be addressed with non-pharmacological approaches. In the current healthcare model, these are delivered by separate and often disjointed specialist services. This leaves the patients feeling lost in a fragmented healthcare system where clinical outcomes remain suboptimal even with the best current practice applied in each discipline. Our review aims to address this challenge calling for a paradigm change to conceptualise difficult-to-treat asthma as a multimorbid condition of a “Difficult Breathing Syndrome” that consequently needs a holistic personalised care attitude by combining pharmacotherapy with the non-pharmacological approaches. Therefore, we propose a roadmap for an evidence-based multi-disciplinary stepped care model to deliver this.

## 1. Introduction


**The challenging case of Mrs T: when the best clinical practice fails to lead to the best clinical outcomes for a patient with difficult-to-treat asthma.**



*Mrs T is a 48-year-old lady, living with severe asthma and multiple other major health conditions, a constellation often described as difficult-to-treat or problematic asthma. She was referred to our tertiary severe asthma centre in 2016 with severe wheeze. She had been diagnosed with asthma at the age of 12 years but remained mainly symptom free until aged 46. Mrs T was married, had a teenage son with a learning disability, and used to work as a nurse but recently had to stop working due to her problematic asthma. At the time of her referral, apart from asthma, Mrs T already had been diagnosed with depression, obstructive sleep apnoea (OSA) and rhinitis. She was also obese with a high health risk Body Mass Index (BMI) of 40.8 and was struggling to lose weight.*



*Mrs T had been referred to, assessed and treated by many specialist physicians, in a tertiary centre, in her local general hospital, in the nearest community hospital and also received care from her GP. As a result of these assessments, she was diagnosed with gastro-oesophageal reflux disorder (GORD) and obstructive sleep apnoea (OSA). Over the years, she had also been diagnosed with inducible laryngeal obstruction (ILO). At times, she received pharmacological treatment (including oral corticosteroids for her breathing difficulties) based on her symptoms, even when underlying physiological biomarkers remained low. Yet, her condition had worsened. She developed type 2 diabetes and binge eating disorder. These comorbidities have all been treated separately, many without major improvement. For years, Mrs T has been taking at least 15 different medications in form of daily tablets, inhalers, sprays and drops. In addition to optimisation of her asthma treatment plan, a multidisciplinary approach using additional non-pharmacological strategies was also adopted to support her holistic healthcare needs. For a while she has been able to control her asthma better, her OSA improved, she lost significant weight and increased her physical activity. Some attempts have been made to reduce the number of medications but these deemed unsuccessful in the long run.*


In this paper we address the challenges in the multidisciplinary management of difficult-to-treat asthma in adults like in the case of Mrs T and review the non-pharmacological approaches that, although acknowledged as being effective at addressing some treatable traits, are still underutilised in clinical practice. We propose the reconceptualising of difficult-to-treat asthma as a multimorbid condition, that might be best regarded as a “Difficult Breathing Syndrome” (DBS) and, based on the emerging evidence from multimorbidity research, we put forward easily implementable and cost-effective management approaches that offer the potential to meet the holistic management needs of DBS.

### 1.1. Difficult-to-Treat Asthma in Context

Asthma is a common chronic inflammatory airway disease which is estimated to affect over 300 million people globally across the life course [1]. Most people with mild to moderate asthma obtain good disease control of the typical associated symptoms of breathlessness, chest tightness, wheeze and cough with inhaled therapies following standard guideline-based approaches [2]. However, around 3–10% of people with asthma have a more severe/difficult-to-treat (or difficult) disease [3] with greater disease morbidity, healthcare dependency, treatment needs and potential mortality risk. While comprising a small fraction of the asthma population, such patients account for a significant proportion of the burden associated with asthma, accounting for more than 60% of asthma-associated healthcare costs [3,4]. As a result, there has been a concerted effort in recent years to better understand the nature and driving mechanisms behind more problematic asthma informing the development of effective treatments that target the relevant airway pathophysiology.

Current pharmacotherapeutic approaches to asthma are moulded to the type 2 (T2) inflammation paradigm of asthma pathophysiology. “T2-high” and “T2-low” asthma inflammatory endotypes defined by the presence or absence of T2 inflammatory processes have become the central framework for currently classifying asthma pathophysiology [5]. In parallel, the last 5 years has seen a rapid expansion of T2-targeting higher level anti-inflammatory biologic asthma treatments that have entered clinical practice globally. Agents such as Omalizumab, Mepolizumab, Reslizumab, Benralizumab and Dupilumab have undoubtedly delivered improvements in patient outcomes. Yet not all patients respond well to biologic treatments [6] and recent studies have shown that patients within the T2 category of disease experience the most problematic symptoms [7,8]. To understand the heterogeneity of clinical presentations and treatment responses seen in patients with difficult asthma therefore requires looking beyond the pathophysiological understanding of the T2 paradigm.

As understanding of the pathophysiology of more severe asthma has grown, so has the recognition of the “real-world context”—that problematic asthma is often part of a wider complex web of adverse health issues. Discrete definitions for “difficult asthma” and “severe asthma” exist as outlined by the Global Initiative for the Management of Asthma (GINA) [9]. As such, difficult asthma describes asthma where aggravating comorbidities, inadequate treatment, suboptimal inhaler technique and/or poor adherence may individually or collectively contribute to poor asthma control. Within this wide-ranging spectrum exists a subset of patients with truly severe asthma that remain sub-optimally controlled despite optimised treatment of both asthma and contributory factors [10,11,12,13]. Severe asthma has been defined by the European Respiratory Society/American Thoracic Society as asthma needing treatment with guideline-suggested medications for GINA steps 4–5 asthma (high dose ICS and long acting beta agonist (LABA) or leukotriene modifier/theophylline) for the previous year or systemic steroids for >50% of the previous year to prevent it from becoming “uncontrolled” or remaining “uncontrolled” despite this therapy [14]. With the comprehensive assessment of patients with problematic asthma, it is evident that most fall into the category of difficult rather than severe asthma. In two recent European studies, between 12 and 20% of patients with difficult asthma were diagnosed with severe asthma [15,16].

### 1.2. Difficult Asthma as Part of a Multimorbidity Difficult Breathing Syndrome—The Concept of Treatable Traits

It is now well recognised that many patients with asthma do not attain good asthma control despite full optimisation with currently available asthma treatments. [17]. In addition, it is increasingly apparent that at the more “difficult-to-control” end of the spectrum, asthma often constitutes part of a multimorbidity constellation of conditions that contribute to the burden of asthma and is better regarded as a “Difficult Breathing Syndrome” (DBS) rather than “severe asthma” alone. This holistic stance better reflects the numerous challenges faced by patients with problematic asthma (Figure 1).

An important new taxonomic approach to airways disease based on identifying and managing component factors rather than generic disease labels such as asthma was recently proposed by Augusti et al. to provide structure to the understanding of multimorbidity in airways diseases like difficult asthma [18]. Thereby, potentially modifiable factors, known as “treatable traits”, are broadly categorised as pulmonary, extrapulmonary, behavioural and environmental. These are acknowledged to potentially occur in combinations that are specific to an individual patient. Pulmonary traits might include fixed airflow limitation, small airways disease, pattern of airway inflammation (eosinophilic, neutrophilic, mixed inflammatory, paucicellular), allergic fungal airways disease, aspirin exacerbated respiratory disease, bronchiectasis, airway infections and dual COPD [18]. Extrapulmonary traits may be both physical or psychophysiological. Physical comorbidities include rhinitis, chronic rhinosinusitis (with or without polyps), gastro-oesophageal reflux disease, obesity, obstructive sleep apnoea (OSA) and physical deconditioning [18]. Psychophysiological comorbidities include breathing pattern disorder, inducible laryngeal obstruction (ILO), anxiety and depression. Behavioural and environmental traits include poor inhaler technique, poor adherence to treatment, distorted symptom perception and exposure to smoking (active and passive) and pollution. A core goal of this systematic identification of treatable traits is to highlight the underlying complexity of clinical presentation to facilitate more targeted and holistic approaches beyond pure asthma management at an individual patient level [18]. This is a notable shift from the “one size fits all” approach encouraged by traditional guideline-based management strategies that have been the mainstay of asthma treatment in recent decades.

While such understanding of the components of difficult asthma highlights the presence and role of individual comorbidities, it also raises a less well-recognised concept: multimorbidity. Treatable traits are common in difficult asthma where they may congregate together in varying combinations within individual patients [19,20,21]. Multimorbidity can be described as three or more co-existing morbidities, that may include physical or psychological diseases, where none of the conditions is more central than the others, as opposed to comorbidities that accompany an index disease [22]. Multimorbid conditions can but do not necessarily share biological or aetiological links [23]. A recent overview of 53 systematic reviews in multimorbidity [24] identified three main disease patterns: cardiovascular and metabolic disease, mental health-related problems and musculoskeletal disorders. The most common diseases in multimorbidity were cancer, hypertension, heart disease, stroke, diabetes, arthritis/osteoarthritis, osteoporosis, depression, COPD and asthma. Promising multimorbidity management pilot programmes have been developed for diabetes and cancer in several European countries [25] that may also be translated into patients with difficult asthma. However, the notion of multimorbidity in difficult asthma and how that impacts individual patient outcomes has attracted limited focus in the literature compared to comorbidity. A recent study found a median number of three comorbidities per patient attending a specialist-referral difficult asthma clinic in Melbourne, Australia [26]. Importantly, the burden of treatable traits appears to align with worse asthma outcomes such as exacerbations, asthma control and quality of life. Conversely, systematic clinical approaches that incorporate addressing treatable traits in asthma management have also recently shown clinical effectiveness in improving outcomes for this patient group [27]. This model of difficult asthma as a DBS with multiple treatable traits further stimulates the need to manage such patients using multi-disciplinary approaches based on individual patient needs. Real-world studies clearly demonstrate the significant level of ongoing comorbidity and potential multimorbidity seen in patients with difficult asthma. For example, in the Wessex AsThma CoHort of difficult asthma (WATCH) study based in the tertiary referral Difficult Asthma Clinic at Southampton, United Kingdom (UK) [28], a high prevalence of physical comorbidities like rhinitis, gastro-oesophageal reflux disease (GORD) and obesity were observed. However, so too were psychophysiologic comorbidities such as anxiety, depression, breathing pattern disorder patterns and inducible laryngeal obstruction/vocal cord dysfunction (Figure 2). Recent findings from the WATCH study have also demonstrated differing associations of these various comorbidities with difficult asthma phenotypes based on age of asthma onset/sex which merit a wider understanding [29]. Specifically, psychophysiologic comorbidities and obesity tended to be more common in females with difficult asthma in that study highlighting other treatment options beyond asthma pharmacotherapy for particular subgroups [29].

### 1.3. Structured Multi-Disciplinary Team Approaches to Difficult Asthma Care

The growing portfolio of higher-level biologic medications alongside recognition of a multimorbidity model with numerous treatable traits for difficult asthma has fuelled adoption of increasingly structured approaches to care for patients with difficult asthma [30]. A key aim of such approaches is to both address the asthmatic disease process as well as relevant aggravating comorbidities in such patients. That in turn has been accompanied by an increasing focus on multidisciplinary team (MDT) models of care that collectively support the diverse care needs of this patient group. Such structured models of care will inevitably vary according to the healthcare system and available resources. This structured approach lends itself particularly well to implementation via specialist care centres for patients with difficult asthma. In countries such as the UK, this approach has been further aligned to a process of regional specialist centres for difficult asthma supporting regional networks of care [30]. These centres must meet specified resource requirements and are subject to quality benchmarking on core outcomes. While the UK specialist commissioned framework offers one systematic approach, data has consistently shown that comprehensive assessment within more specialised difficult asthma care realises improvements in patient asthma status regardless of the geography or healthcare system [31]. Thus a three-step systematic approach to difficult asthma specialist care based on diagnostic confirmation, comorbidity detection and inflammatory phenotyping was assessed in Melbourne, Australia. This resulted in significant improvements in comorbid conditions like chronic rhinosinusitis and breathing pattern disorder. It also resulted in significant parallel improvements in asthma-related outcomes such as asthma control, asthma-related quality of life and exacerbation frequency. Further work from the same research group has more closely focused on asthma patient-related outcome measures [32]. This found that a systematic assessment framework in difficult asthma specialist care realized significant improvements across multiple asthma domains. These included a halving of maintenance oral corticosteroid dose (regardless of biologic co-administration) and achievement of minimally important differences for asthma symptom control and quality of life in over 50% patients. Reduced exacerbations were found in 64% patients while 40% patients improved their FEV1 by ≥100 mL. Improvement in at least one domain was found in 87% of patients undergoing that systematic assessment. Of note, the improvements demonstrated in this study were independent of biologic treatment initiation, highlighting the value of early adoption of such approaches in the patient care pathway to ensure focusing the right treatments on the right patients, at the right time. In that context, structured assessment can be applied at different points along the asthma care pathway, not just in a specialist centre environment. SIMPLES was introduced as a tool for use in primary care to support management of patients with poorly controlled asthma [33]. Smoking status, Inhaler technique, Monitoring, Pharmacotherapy, Lifestyle, Education and Support (SIMPLES) assessment was introduced as a tool for use in primary care to support management of patients with poorly controlled asthma [33]. It encompassed self-management, education, monitoring and lifestyle changes (with emphasis on smoking status) in addition to pharmacotherapy. Often ignored facets such as regular review and accessibility were also recognised and given prominence. This was coupled to guidance on when to refer from primary to specialist care. Another important component to SIMPLES was the early adoption of digital technologies with web-based access to both the SIMPLES framework and relevant assessment tools. More recently the Severe Asthma Toolkit was developed as a holistic resource to support structured multidisciplinary care for patients with severe asthma across the healthcare spectrum [34]. Developed by a consortium of multidisciplinary experts with patient and advocate codesign, this resource was established in the format of an easily accessible website. Content included background information about severe asthma, diagnosis and assessment, management, medications, comorbidities, living with severe asthma, information on establishing a clinical service, specifics to paediatric and adolescent care, advice on specific population needs, registries and access to relevant supporting resources [34].

### 1.4. The MDT Components of Specialist Difficult Asthma Care

In a specialist difficult asthma clinic setting, the assembled MDT typically includes a range of healthcare professionals (HCP) including consultant respiratory physicians, consultant allergists, asthma nurse specialists, asthma physiotherapists, asthma psychologists, asthma pharmacists, speech and language therapists and dietitians [35,36,37]. Patients referred to such services will generally undergo a comprehensive assessment at the point of referral followed by an appropriate pharmacotherapeutic treatment optimisation [30]. They then have regular follow-ups with appropriate members of the MDT as dictated by the individual's need. In parallel they may receive non-pharmacological treatment measures to address relevant comorbidities typically under the guidance of allied health professionals. Such MDTs review cases on a regular (often weekly) basis in a meeting setting to achieve group consensus on appropriate treatment steps culminating in approval for higher level biologic treatments once the MDT is satisfied that other appropriate actions have been addressed. This structured pathway meets the important goal of ensuring that all other facets of patient needs are met rather than simply escalating to higher asthma therapies in the hope of improving refractory breathing difficulties.

Pharmacological approaches in the treatment of difficult asthma and associated comorbidities are well reported in the literature, but as the number of coexisting morbidities increase so does the iatrogenic risk from polypharmacy. Denton et al. carried out a protocolised systematic assessment to identify and manage factors that contribute to difficult-to-treat asthma leading to holistic management approaches combining pharmacological and non-pharmacological treatment components. Over a 6-month follow-up period, this approach led to 87% of patients improving in at least one asthma outcome despite halving their oral corticosteroid use. This was independent of their monoclonal therapy and was equivalent improvement to what can be seen with monoclonal biological therapy [32]. This result suggests that a stratified approach combining pharmacological and non-pharmacological treatment options may be advantageous in DBS. The appropriate treatment option for the particular treatable trait will need to be based on a holistic needs and risk assessment of the patient.

The benefits of non-pharmacological approaches in multimorbid conditions are increasingly recognised, however evidence is still sparse: as recently as in 2014, Kenning et al. observed that 60% of studies they reviewed had excluded participants with multimorbidity [23]. Since then, the number of research studies including participants living with multimorbidity has been growing but reviews on effectiveness of interventions to improve multimorbidity outcomes are often inconclusive, owing to the heterogeneity of approaches and settings [23,38]. Hence, efforts of specifically addressing multimorbidity through non-pharmacological approaches are in their infancy and the reporting of well-designed randomized, controlled trials still needs improvement [39]. In the next part of this paper, we review the non-pharmacological approaches that may be of particular clinical value in addressing key extrapulmonary and behavioural treatable traits within the DBS model of difficult asthma.

## 2. Review Methods

We performed a narrative review of the literature between 2000–2022. Electronic searches included MEDLINE, EMBASE, CINAHL, AMED, PsychINFO, Cochrane Airways Group Centralised Register, EMCARE, PubMed, PsychARTICLES, Francis and Taylor online, Elsevier, ScienceDirect, Sage, Google Scholar and EBSCOhost. Additionally, a hand search of *Journal of Multimorbidity and Comorbidity* was performed. Search terms for primary disease included asthma, chronic obstructive pulmonary, chronic lung disease, respiratory, disease, respiratory disease, respiratory illness. Secondarily, we included searches for the extrapulmonary treatable traits of asthma, such as vocal cord dysfunction, anxiety, depression, psychological comorbidities, psychological distress, psychological dysfunction, breathing pattern disorder, inducible laryngeal obstruction, obesity, gastro-oesophageal reflux disease or GORD and (non-)adherence. We searched specifically for non-pharmacological interventions in extrapulmonary treatable traits in asthma, including terms: non-pharmacological, exercise, dietary, pulmonary rehabilitation, surg*(ical), smoking cessation and psychoeducation. We excluded papers with a non-adult population and papers that included chemical agents as we considered these as other pharmacological interventions. All reviewed papers were published in English.

## 3. Non-Pharmacological Approaches to Extrapulmonary and Behavioural Traits within the Difficult Breathing Syndrome

### 3.1. Biopsychosocial Processes in DBS

Depending on the focus of intervention, non-pharmacological approaches to address treatable traits within a DBS scenario broadly fall into three categories: emotion focused, predominantly addressing psychological distress; behavioural, aiming to support behaviour change for optimised self-management; and educational, to increase patients’ knowledge and understanding of the illness and equip them with skills for self-management. In reality, interventions often combine two or all of these aspects. Indeed, patients’ adaptation to, and self-management of, DBS requires a trinity of interlinked emotional, cognitive and behavioural components. Yii and Koch’s framework [40] offers a simple overview of how these components interact (Figure 3). Examples are given to illustrate these processes within the context of DBS also highlighting the two-way interactions between emotions, cognitions (beliefs, perceptions, thoughts, attitudes) and behaviours. In this, a treatable trait might be the cause or effect of other treatable traits.

The associated physiological processes are less well understood but evidence is emerging that a number of shared biomarkers are likely to be implicated in the neuroimmunological and neuroendocrine systems as in several comorbidities in the DBS. Studies found that both depressed and anxious non-asthmatic patients had higher levels of C-reactive protein (CRP), interleukin (lL)-6, tumour necrosis factor-alpha (TNF-α), interferon-gamma (IFN-ϒ) than healthy controls [41,42] and depressive symptom-associated IL-1β and TNF-α release correlated with impaired bronchodilator response and neutrophilic airway inflammation in asthma [43,44]. In our Wessex Severe Asthma Cohort study (Varkonyi-Sepp et al. 2022, unpublished), an examination of the molecular associates with overall levels of psychological distress revealed a complex network of pro-inflammatory protein markers associated with high overall distress. Additionally, higher blood and sputum neutrophil count and CRP were associated with higher distress levels, pointing to the role of inflammation in impaired mood states. A recent review by Roohi et al. [45] showed the role of IL-6 in the biological processes associated with depression and a group in Finland reported elevated serum levels of IL-5 are associated with an increased likelihood of depressive disorders, suggesting a number of possible pathways. Central nervous system and cortex alterations both in hypersensitisation to inflammatory and hypoxic environment and morphology have also been described as possible mechanisms for psychological distress, altered symptom perception and illness behaviour for example suboptimal healthcare use or breathing pattern disorder/breathing pattern disorder [39,40,41,42,45,46,47].

#### 3.1.1. Emotion-Focused Approaches

Many treatable traits in the DBS (for example obesity, chronic rhinosinusitis, breathing pattern disorder, GORD, inducible laryngeal obstruction) are independently associated with higher levels of psychological distress with anxiety and depression being the most often measured ones across research studies [12,46,47,48,49]. The psychological impact of living with DBS has not been established but it is known that difficult asthma and comorbidities are associated with a higher prevalence of psychological distress [12,49,50,51]. Furthermore, in general, the prevalence of depression is twice as high in individuals with multimorbidity than in those without and the incidence of anxiety, depression and stress are positively correlated with the number of morbidities in multimorbidity [52,53].

Anxiety and depression are the most common mood disorders in asthma. Studies have described substantial prevalence of both anxiety (13–80%) and depression (16–59%) in patients with severe uncontrolled asthma which might be dependent on comorbidities [21,50,54,55,56]. Anxiety and depression differ in possible causes, presentation and how they might impact asthma outcomes, but they can also present together. A large range of non-pharmacological emotion-focused interventions to alleviate anxiety and depression in people with asthma were trialled extensively, but widespread variations in the outcomes, measurement, and the intervention design and delivery mean that their effectiveness remains inconclusive [57,58,59]. The most promising approaches so far have been cognitive behavioural therapy (CBT) [60], counselling and various forms of third-wave interventions such as mindfulness [12,59].

CBT is a type of psychotherapy. Psychotherapies are non-pharmacological treatments in which the therapist and patient(s) work together to address psychological conditions and/or functional impairment using psychological models and techniques. They may focus on any combination of the patient’s attitudes, thoughts, affect, and behaviour, social context and development [61]. CBT aims to identify distorted and often automatic negative thoughts and to challenge these and the underlying dysfunctional beliefs. It also uses behavioural tasks of diary-keeping and validity-testing of beliefs between sessions alongside skills training within the therapy sessions. Sometimes cognitive and behavioural therapies are also used separately.

Relaxation approaches include progressive relaxation (systematically tensing and relaxing large muscle groups in different parts of the body), autogenic training (learning to create relaxed state by mentally controlling bodily sensations), hypnosis (deep relaxation that may be induced using mental imagery, often accompanied by autosuggestion to create positive thoughts and feelings), as well as third-wave treatments such as mindfulness and acceptance and commitment therapy, that is learning to be aware of internal and external occurrences in the present moment and accepting these without taking mental or physical action [62]. Biofeedback, a treatment approach to alleviate anxiety and stress, can also be considered a behavioural intervention. Biofeedback can use a variety of sensors to measure breathing, heart rate and skin electricity, and with the help of these objective measures, the person learns to recognise physiological signals of mood states and then applies relaxation strategies to control these, with feedback from the sensors. Because the equipment used for this approach is costly and setting up the treatment is complicated, this approach is less used than other relaxation methods.

Counselling is talking over problems with a health professional. It is often less structured and shorter than psychotherapy and tends to focus on the problems in the present.

Whilst there is evidence that improved symptoms of depression and anxiety are associated with a better self-report of asthma control [63], the direction of this relationship is unclear. In difficult asthma, self-management requires carrying out complex sets of behaviour changes. For example, this includes adhering to the treatment regime, monitoring one’s environment to avoid triggers, monitoring symptoms, diet and/or physical activity and taking necessary actions if their asthma worsens. This puts a high demand on people’s cognitive and behavioural capacity that often contributes to their high levels of psychological distress [57]. It is reasonable to assume that in a DBS setting with multimorbidity this impact is even higher [12]. The combination of comorbidities that compose a DBS can greatly vary. This poses treatment challenges and requires personalised approaches depending on what morbidities make up the individual person’s DBS.

Moreover, many patients do not present with clinical levels of distress but could benefit from improved knowledge about the different elements of their DBS (i.e., addressing the cognitive psychological component for example through psychoeducation) and from support to gain behavioural skills and competence to self-manage [46,64]. Evidence suggests that psychoeducation including symptom recognition, inhaler technique and biofeedback and behavioural interventions such as medication adherence, breathing retraining, exercise, losing weight, avoiding triggers and stopping smoking may improve clinical outcomes of these treatable traits separately. Some, like breathing training, can improve voluntary control and modification of symptoms, others help by developing strategies for improved self-management, for example medication adherence or through changing lifestyles [12,58,59].

#### 3.1.2. Behavioural Approaches

Patients who have asthma and are obese are now considered a distinct cluster of patients with poorer asthma control, reduced treatment responsiveness, and increased asthma severity and exacerbation rates [47]. The prevalence of obesity in difficult and severe asthma have been demonstrated to be as high as 47% [12]. Patients who are obese and have difficult asthma demonstrate poorer symptom control, higher levels of comorbid anxiety and depression and lose a higher number of days to illness than non-obese difficult asthma patients. Within obese asthma patients, comorbidities are common, and many of the comorbidities contributing to sub-optimal control in difficult asthma are also associated with obesity [49]. The mechanistic link between asthma and obesity requires further clarification and is likely multifactorial. There is suggestion of at least two phenotypes of obese asthma with early onset atopic obese asthma and late onset non-atopic asthma [48]. The pathophysiology between early onset atopic obese asthma and late onset non-atopic asthma may vary because the latter resolves after weight loss [65].

There is also a mechanic effect of obesity in asthma, in that adipose tissue around the upper airway results in dyspnoea due to upper airway narrowing, and limits breathing with a reduction in vital capacity and increased respiratory resistance, resulting in a restrictive lung deficit [50]. In addition, there is chronic inflammation associated with obesity.

Weight loss interventions to address obesity in asthma can target diet (to reduce calorie intake and/or improve nutritional value of food), physical activity or a combination of both. There is evidence that a ~10% weight loss on its own improves asthma outcomes [66,67,68]. Dietary changes alone, whether prescribed by dietician or complemented by diet-focused counselling, often showed little effect on a variety of asthma outcomes [69] but improvements were seen when dietary interventions were combined with physical activity programmes. [70,71]. An evaluation of the long-term maintenance of these behaviours and their benefits however, is lacking.

The link between exercise and asthma control is now well established, with exercise interventions demonstrated to improve quality of life, lung function and symptom scores [40]. Additionally, recent small studies suggest there may be a reduction in systemic inflammation [41] and improvement in redox buffering capacity [42], with the suggestion that this may contribute to the mechanism through which the clinical benefits are conveyed. Even moderate levels of exercise showed an effect on eosinophilic asthma airway inflammation [72]. Exercise has also been shown to impact on many of the comorbidities or treatable traits commonly seen with asthma, and treatable traits are increasingly appreciated to contribute towards disease control [43]. Combined aerobic and resistance exercise intervention have shown efficacy on multiple asthma outcomes [57,58]. A wide variety of behavioural approaches to promoting physical activity (PA) in people living with asthma have been trialled, as reviewed by Tyson et al. [73]. Some aimed to reduce sedentary time and others to increase time spent being physically active. Interventions ranged from supervised aerobic exercise sessions to unsupervised walking. Frequency of sessions was mostly between 1–3/week and the intervention period varied between up to 3 and 12 months. A number of interventions used contracting, healthcare professional prescription and/or material incentives to increase engagement with the programme. Some interventions combined physical activity and weight loss intervention. Settings also varied as well as the practitioner delivering the intervention. A large range of behaviour change techniques were used, but the review highlighted that they did not include techniques that help self-regulated behaviour and sustained motivation, essential for adopting and maintaining behaviour change. Interventions led to the increase of PA during the intervention period, but only one study included information on follow up and this reported that the intervention effect was not maintained [74]. In individuals living with multimorbidity, exercise was deemed safe and led to improvements in health-related quality of life and depression. The effect was more pronounced in younger people and those with higher baseline depression levels, however the evidence was weak and therefore more research is needed in this population [75].

Non-pharmacological approaches for smoking cessation include behavioural support, especially the use of communication strategies aiming to increase motivation. Even very brief conversations were shown to be effective when they used open-ended questions, reflective listening and summarizing, and if an initial conversation is followed up that is even more useful [76]. Patient perceptions about the benefits and disadvantages of quitting and also about barriers and facilitators can be elicited and then appropriate behavioural strategies, including setting specific plans for trigger and high-risk situations, can be developed and, if possible, periodically reviewed [77,78,79]. Sometimes these approaches are combined with elements of psychological therapies to control thoughts urging to smoke. Nicotine replacement can complement these non-pharmacological components.

To address breathing pattern disorders, in reviews of a number of breathing retraining approaches [80,81,82], including the Buteyko breathing technique, yoga and inspiratory muscle training, provided some improvement in asthma control outcomes [83] and there is evidence on breathing retraining leading to improvements in quality of life independent of effects on lung function or airway inflammation [84], but it is yet to be evaluated in patients living with DBS.

#### 3.1.3. Educational Approaches

Psychoeducation recognises the importance of sufficient asthma-related knowledge, of patients’ beliefs about the nature of their illness and the treatment (illness perceptions) as well as the role of patients’ confidence (self-efficacy) [85] to carry out the required tasks to manage their condition.

An important contributor to treatment adherence is the set of beliefs, so called illness perceptions, about the efficacy or necessity of the medication or treatment plan, the treatability of the condition and about the illness in itself [86,87]. These beliefs also impact on behavioural adherence for example avoiding dealing with asthma and dismissing medical advice or consciously and consistently self-managing. Moreover, they are often linked with patients’ understanding and knowledge about the condition and the treatments [88,89] that consequently impact on their self-confidence to manage their condition (so called self-efficacy), including inhaler use and treatment adherence [90,91]. Patients reported that the lack of self-efficacy to recognise their asthma symptoms and also to distinguish these accurately from other symptoms, such as breathlessness caused by physical exertion, prevented them from the appropriate use of medication and from avoiding triggers [92]. Depending on the patient’s level of knowledge, self-efficacy to self-manage and their illness perception, a stratified combination of asthma education with checking correct inhaler technique, good HCP-patient relationship and interventions to increase patients’ motivation to self-manage, can be beneficial to improve treatment adherence [92,93]. Indeed, written or oral asthma action plans, that is detailed plans for self-management including actions when asthma worsens, also showed some benefits, but further research is needed to fully understand their most effective components, their implementation barriers and facilitators both from patients’ and healthcare professionals’ points of view [94,95].

Psychoeducational interventions alone have led to modest and short-term improvements in self-management and in reducing hospital admissions but they were effective mainly in patients with single morbidity thus it is unclear whether this approach would make a difference in people with DBS [96]. A combined psychoeducation and emotional, cognitive and behavioural self-regulation intervention [97] led to improvements in quality of life, asthma symptoms, treatment adherence, peak expiratory flow rate, asthma-related knowledge, attitude towards asthma, self-efficacy, and negative emotionality immediately after the programme interventions and were also sustained at a 3 month follow-up. The study group however involved patients with mild to moderate asthma thus these results should be validated in groups of people with severe multi-morbid asthma as seen in DBS. Conversely, a systematic review by Smith et al. [98] concluded that psychoeducational interventions did not lead to a sustained improvement in outcomes in reducing hospitalisation, improving quality of life or psychological comorbidities in patients whose multimorbidity included asthma, although the 17 studies reviewed included a wide variety of settings, delivery methods and objectives.

#### 3.1.4. Interventions Combining Emotion-Focused, Behavioural and Educational Approaches

Inducible laryngeal obstruction (ILO) is a disorder of the laryngeal area [99,100,101,102]. In their join statement, the European Respiratory Society and European Laryngological Society recognise the role of psychological causes in certain cases of ILO [103]. Some brain regions implicated in laryngeal hyperresponsiveness, a cause of ILO, are also reactive to stress and emotions [99]. ILO is often associated with psychiatric conditions [102,104] for example major depression, anxiety disorders and somatoform disorders (in which patients express psychological stress in physical symptoms often caused by inability to express their emotions directly). Whilst a purely psychiatric aetiology of ILO is debated, it is plausible that complex inflammatory processes are at play as suggested in the case of a number of medically unexplained symptoms by Hyland at al [105]. This is particularly relevant considering the role of both systemic and local inflammation in asthma and indeed, in several comorbidities in the DBS, including mood disorders. In the treatment of ILO, a number of authors [100,101] suggested a combined intervention addressing the cognitive (psychoeducation about the pathophysiology and behavioural aspects of the dysfunction), behavioural (breathing techniques to relax the larynx, strategies to resist the urge to clear throat or cough, biofeedback) and emotional (psychological counselling, hypnotherapy) components of this complex condition. Papers describing such interventions emphasize the idiosyncratic nature of symptom presentation and, consequently, the importance of personalised approaches in applying the components. This combined intervention is described as being implemented in clinical practice however objective evaluation of its effectiveness is lacking. Breathing retraining is often applied to improve ILO, and it is frequently combined with psychoeducation on breathing pattern disorder [64] and speech therapy [102].

A promising approach that recognises the complexity of breathlessness and aims to address its emotional, cognitive and behavioural aspects is the ‘Breathing, Thinking, Functioning’ evidence-based approach that showed efficacy in chronic obstructive respiratory disease. It is suggested to be applicable to advanced cardio-respiratory conditions, however it has not been trialled in severe asthma [106]. As described above, alongside psychoeducation, there is also a role for behavioural interventions to support treatment adherence for example with the use of reminders, personalised messaging or visual triggers as a reminder to take the medication [90]. Medication adherence and inhaler technique however has been negatively associated with self-management in older adults with low literacy [107]. Multimorbidity increases patients’ treatment burden and might lead to impaired treatment adherence. [108]. Maffoni et al. [109] identified several factors that influence treatment adherence in multimorbidity such as patients’ beliefs about polypharmacy and drug prioritization, patient’s experience and capabilities, the prescriber–patient relationship, health literacy, treatment characteristics and complexity, and family and social support, some of which we also discussed in the educational approaches. Hall et al. proposed a multi-component pulmonary rehabilitation approach including exercise, a weight-loss programme, self-management approaches, psychoeducation, breathing training and an asthma action plan [110] that might improve patients’ quality of life and reduce side-effects of pharmacological therapy [111]. The components however were not reviewed in a multimorbidity contexts thus further work is needed to assess the efficacy of such an approach in DBS.

#### 3.1.5. Other Non-Pharmacological Approaches

Bariatric surgery might be useful for the long-term treatment of obesity in asthma however evidence is inconclusive and more research is needed [112,113]. The discussion of surgical interventions is beyond the scope of this review, however, it is important to point out that bariatric surgery is best complemented by post-operative behavioural management support to promote maintenance of healthy eating and physical activity habits that are necessary for sustained weight control. In addition, pre-operative psychological screening and pre- and post-operative psychological treatment if necessary is also recommended, as obesity is often associated with psychological distress and/or eating disorders and low-self-esteem leading to impairment of self-confidence to maintain behaviours that prevent weight regain (e.g., food intake and physical activity) [114,115,116].

Surgical interventions have also shown benefits in the treatment of GORD [117] but similarly to bariatric surgery, the best long-term results are achieved when surgery is combined with behavioural support to manage a specific diet that maintains the benefits of the surgery. GORD is also often triggered or aggravated by chronic stress, therefore emotion-focused interventions that equip the person with skills to decrease their stress levels will be also beneficial [118].

Supraglottoplasty, a microscopic surgical procedure to alter structures of the upper larynx, has shown benefits in exercise-induced ILO and in clinically significant breathing problems, however, randomised controlled trials of the procedure are lacking and the long-term benefits have not yet been established [103,119].

Figure 4 depicts the non-pharmacological treatment approaches shown to be most effective for the treatable traits of a DBS model of difficult asthma.

#### 3.1.6. Telehealth

Although not aimed at one particular treatable trait, the spread of technology into everyday life can offer opportunities to support people’s self-management as it allows communication or information sharing between patients and health care providers over a distance, often in real time [120]. Beyond more obvious behaviour change functions like reminders to take medication and/or feedback on exercise and/or nutrition, technology can also provide information on asthma triggers such as pollution or allergens, medical information, for example peak exploratory flow, or advice, for example on emotional self-management or diet.

A Cochrane review concluded that whilst the benefit of telehealth interventions (care provided from a distance using technology—for example telephone, internet, mobile phone, remote monitoring devices) was low for people with mild to moderate asthma, it has shown more of an effect for people living with severe asthma, especially if it was interactive, enhancing the communication between the patient and their clinical team [121]. A recent qualitative study explored patients’ and clinicians’ view of using technology: ‘an internet of things’ (joining up various technology like mobile phone apps, telehealth devices, electronic clinical data, environmental data) for supported self-management of asthma [122]. It indicated a preference for a system that requires minimal input from users and provides real-time advice to help them learn about their asthma, identify and avoid triggers, and allows clinicians to have real-time data about patients’ clinical status, which allows decisions to adjust the patients’ treatment. Patients preferred real-time data on peak flow, environmental triggers (pollen, humidity, air temperature) and asthma symptoms. The clinicians’ preference was data on patients’ conditions that they could access during consultations. Some patients also preferred lifestyle behaviour logs but this was not a priority for clinicians. Telehealth interventions in asthma management promise value but only if the automated functions are complemented by periodic reviews and as-needed consultations with the clinical team [121]. Conversely, a variety of telehealth interventions reviewed across a range of separate health conditions: diabetes, heart failure, asthma, Chronic Obstructive Pulmonary Disease and cancer, did not show consistent marked benefits [120]. This suggests a variety of needs and preferences depending on the health condition and the individual. The benefit of telehealth-supported care in multimorbidity is yet to be established, including patient and clinician preferences and effective components of such management approaches.


**How did we use current best practice in the treatment of Mrs T?**



*Mrs T had been diagnosed with depression before getting referred to us and she has received counselling in the past but has not found it helpful. Her asthma doctor referred her to the Asthma Clinical Psychologist who worked with Mrs T using a cognitive behavioural therapeutic approach. At the time of her referral, Mrs T had a BMI of 40.8. She found it hard to breathe, had OSA and had restricted mobility. Over the next few years, encouraged by her Asthma MDT (doctor, nurses, psychologist and dietician), she joined Weight Watchers, a specialist non-NHS weight loss service that provides a complex behaviour change intervention. Consequently, Mrs T lost 5 stones (32 kgs) and increased her physical activity but gained more than 2 stones (13 kgs) back when her asthma got worse and she developed a binge eating disorder. Additionally, Mrs T restarted smoking because of increasing stress. Her clinician referred her to the hospital smoking cessation services, with limited success. Mrs T also had periods of breathing pattern disorder and was referred for assessment to the Voice Clinic who confirmed a diagnosis of ILO. She subsequently received a course of speech therapy from the MDT Asthma Speech Therapist and breathing training by the MDT Asthma Physiotherapist that she found useful. She reported that her breathing had improved and breathing problem flair ups had less impact on her. At the same time, she received continuous positive airway pressure (CPAP) therapy for her OSA from her community hospital, after referral by her GP. Over time, her sleep improved but with several chest infections over a year causing her asthma to worsen, and thus the improvement was not sustained.*



*The MDT clinical team also explored if Mrs T could benefit from bariatric surgery however with the level of multimorbidity including her binge eating disorder, she was considered too high risk to be eligible for this. Instead, she continued psychological therapy complemented by behavioural support during the periods when she felt able to cope with controlling her eating.*



*Mrs T felt that she has been running around in circles between the different specialist services, felt completely lost in what she said was a fragmented health care system. She was angry that she needed to keep telling her medical history repeatedly to the different specialists who did not seem to have joined the dots. As her condition became more complex, its impact on her life became worse. She found it increasingly challenging to cope with the everyday struggles with housing, debts, quality of living environment, caring for her disabled son and needing to suspend her driving license because of sleep apnoea.*


Each specialist treated Mrs T with the best possible care for each of her particular co-morbid condition, however neither Mrs T’s objective medical outcomes nor her subjective patient-reported outcomes benefited from this approach in the long run.

Mrs T’s story, and indeed many similar clinical stories from our patients living with multiple health conditions including severe asthma, requires reflection on long-established practices and questions whether we need a paradigm shift from the parallel treatment of multiple comorbidities to a personalised multimodal treatment of multimorbidity, including non-pharmacological approaches alongside conventional drug therapies.

## 4. What Can We Learn from Multimorbidity Research for the Clinical Management of People Living with DBS?

In recent years, comprehensive multimorbidity management models combining patient-centered care, support for healthcare professionals and addressing organisational factors (for example applying a robust interdisciplinary team approach or integrated care across the different care provider organisations) gained popularity. Yet, further work is needed to establish what intervention components are effective and work best in which care setting. Furthermore, due to lack of consistent reporting on intervention development and implementation methods, it often remains unclear whether interventions were underpinned by evidence or theoretical models even though the benefits of such an approach has been established [123,124,125,126]. As Mrs T’s case shows, our current model of care with central decision making and care co-ordination by the asthma physician through multiple separate specialist referrals (Table 1) might not lead to the best outcomes, nor might it meet patient needs and preferences, resulting in impaired cooperation and self-management.

Getting the best possible outcomes for individuals living with multimorbidity is heavily reliant on an optimal combination of personal factors (an individuals’ capacity, motivation, and readiness to share responsibility with their care provider), health care professional factors (skills, knowledge, motivation to implement personalised care and navigate the care system in collaboration with the patient) and organisational factors (available resources, accessibility, support within the system) [24,127,128]. A consistent message from the evidence reviewed in this paper is that complexity in the composition of multimorbidities/treatable traits as well as in the variability of emotional, cognitive and behavioural support needs of individuals living with DBS calls for a stratified personalised clinical approach. It is therefore important to assess patients not only for psychological distress but also for their knowledge, skills and confidence to self-manage and their readiness to do so, including their level of motivation to change their behaviour. Moreover, individuals can be supported to gain knowledge, competence and confidence to self-manage their emotions that, as we saw above, may impact on their cognitive and behavioural processes. This approach is underpinned by the concept of self-regulation, an individual’s ability to appropriately respond in the cognitive, affective, and/or behavioural domains in any given situation and context [129]. This is the fundament to the SafeFit trial [130] that is based on a psychological intervention to improve self-regulation of individuals affected by cancer to optimise their psychological and physical functioning. Anecdotal reports from patients as well as practitioners delivering the intervention suggest that this approach is not only acceptable and feasible to them but also increases patients’ confidence in being able to self-regulate their behaviour, cognition and emotion and practitioners’ confidence in their ability to support patients in this.

Considering this, we propose a new model of multimorbidity care for individuals living with DBS that is depicted in Table 2, with an evidence-based methodological roadmap to implementation.

Addressing complex health needs like DBS, calls for complex multi-disciplinary interventions. This is recognised in the UK by the Department of Health comorbidities framework [131] that sets out the principles including inviting patient participation in designing interventions and implementing the National Institute for Health and Care Excellence (NICE) guidelines in clinical practice. Experts in development of complex interventions highlight the fundamental importance of involving stakeholders in this process using evidence-based development methodology including intervention implementation and evaluation such as the person-based approach [132,133,134,135]. The person-based approach provides a template for iterative methods in the planning, optimisation, evaluation and implementation of behavioural health interventions which a wide range of users can engage with. This, in turn might lead to improved health-related outcomes. In our case history, both Mrs T and her health care team could have benefited from this approach, effectively co-creating with Mrs T’s her multidisciplinary care plan that met the holistic management needs of her multimorbid Difficult Breathing Syndrome as well as her personal preferences and enabled the multidisciplinary team to provide the best possible personalised care. Such approaches have been shown to improve outcomes for patients with multimorbidity in primary care and community settings [38] and it is reasonable to assume their benefit in other settings too although additional research has to establish more precisely what works for whom where in the health care system.

The comorbidities framework specifically recommends the Making Every Contact Count (MECC) approach as an exemplar to upskill public service workforce to then deliver complex and personalised interventions. An evidence-based and robustly evaluated intervention developed within the MECC approach is the Healthy Conversation Skills (HCS) [136,137,138,139]. HCS is an effective, person-centered, solution-focused and empowering approach to support individuals to change behaviour to address chronic disease risk factors. It empowers people to identify issues, for example insufficient self-management, and explore barriers then generate solutions to overcome these barriers and make plans for change, followed by review and revision of these plans if needed. HCS addresses many of the critical observations raised in the above reviews, notably that successful strategies should be personalised and should increase people’s self-efficacy to carry out their target behaviour and increase their engagement with and motivation to enact the planned changes and to maintain these long-term. HCS has been shown to increase the confidence and competence of frontline staff to support people to change their behaviour across sectors, professional roles and organisations hence can be used to develop and deliver individualised care plans with complex interventions [138,140,141].

In the UK, the National Health Service's (NHS) long-term plan 2019 [142] lays down the foundation for an integrated care where various care providers working together to meet patients’ individual needs and preferences provide the most cost and resource effective high standard care across systems: a stepped care model [143]. This would allow a joined-up multi-disciplinary, multi-agency pathway to offer a seamless journey to Mrs T where she would not feel lost in the system like so many patients feel within the current traditional care pathways [25,38,144]. The fundamental role of organisational factors (collaboration across teams, organisations, parts of the health care system) has been consistently highlighted in the evidence base [22,24,38,144]. Mrs T would ideally have a care coordinator as her main point of contact who helps her navigate the system and who also provides advocacy with the various health care professionals [38]. The role should be assumed by a professional based on patient needs, for example a psychological wellbeing practitioner for individuals with psychological and physical multimorbidity where addressing depression was the main focus [145] or an occupational therapist [146] or social prescriber or health coach in primary care and in the community [147,148].

This approach could also offer health economic benefits. On one hand, it can decrease direct healthcare cost through improved clinical outcomes resulting from better self-management and reduced overuse of healthcare through better self-regulation and improved symptom perception. Conversely, co-creating personalised care plans can improve concordance and thus can reduce healthcare waste (e.g., incorrect inhaler use, not taking medication, unnecessary/unwanted psychological treatment) [149]. Moreover, as we see in the SafeFit trial [130], behaviour change support and education can be delivered by appropriately upskilled healthcare or wellbeing professionals (social prescribers, health coaches, personal trainers trained to work with people with health conditions) reducing demand on a highly specialist and expensive workforce. Digital technology offers promises in supporting personalised, cost-effective care, although effect sizes have varied and comparatively few have been tailored for the complex needs of severe or difficult asthma [150].

## 5. Conclusions

Asthma control for many patients remains poor despite full optimisation with currently available asthma treatments. Especially at the “difficult-to-control” end of the spectrum, asthma is often accompanied by a variable number of comorbidities increasing the burden of asthma. We called this framework a “Difficult Breathing Syndrome” (DBS) and suggested to adopt a more holistic viewpoint and approach to understand the numerous challenges faced by patients with problematic asthma. Many comorbidities, called ‘treatable traits’ can be treated by pharmacological agents but with the added risk of polypharmacy. Whilst pharmacological treatment is irreplaceable in addressing pulmonary traits, many (particularly extrapulmonary) treatable traits can best be addressed by non-pharmacological interventions that we reviewed in this paper. These, applied alongside newer biological therapies, could augment responses to those biological treatments and thus maximise patients' benefits. In the current healthcare model, extrapulmonary treatable traits are addressed with separate and often disjointed specialist services that provide non-pharmacological treatments. Even though they individually deliver to the current best practice described in this review, the fragmented nature of this model leaves patients feeling lost in the system and hinders achieving the best outcomes for those living with a Difficult Breathing Syndrome (DBS). In this review, we propose a paradigm change from viewing DBS as an asthma condition with comorbidities to conceptualising it as a multimorbid condition managed under asthma services. This approach involves patients and their clinician co-developing a personalised care plan and a needs and risk-based management with options for pharmacological and non-pharmacological treatment delivered by the multi-disciplinary team across systems in a stepped care model. Learning from the emerging multimorbidity research field, we proposed a new care model and provided an evidence-based roadmap to achieving this. Optimally addressing the DBS in everyday clinical practice will not be without challenges in often resource-restricted healthcare systems. It will require a cultural change that values a holistic patient-centred approach and which recognises that investment in coordinated multidisciplinary care pathways is essential, not just desirable. Future research needs to generate more understanding and robust evidence to offer the best personalised treatment and care models for people living with asthma as one in many factors of a “Difficult Breathing Syndrome”.

## Figures and Tables

**Figure 1 jpm-12-01435-f001:**
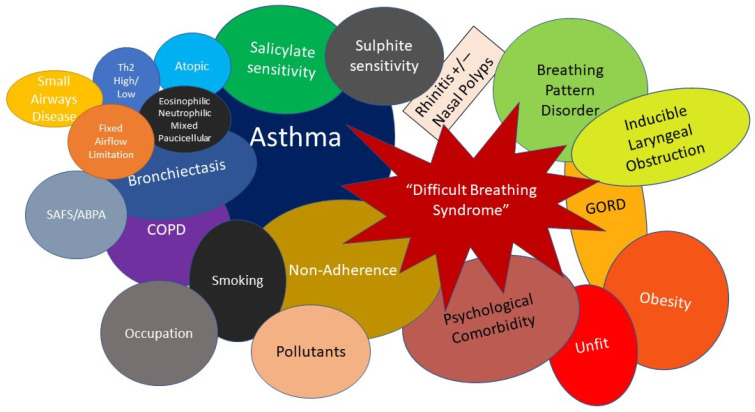
Schematic diagram of the “Difficult Breathing Syndrome” in difficult asthma. T2—Type 2 inflammation, ABPA—Allergic Bronchopulmonary Aspergillosis, SAFS—Severe Asthma with Fungal Sensitisation, COPD—Chronic Obstructive Pulmonary Disease, GORD—Gastro-oesophageal reflux disease.

**Figure 2 jpm-12-01435-f002:**
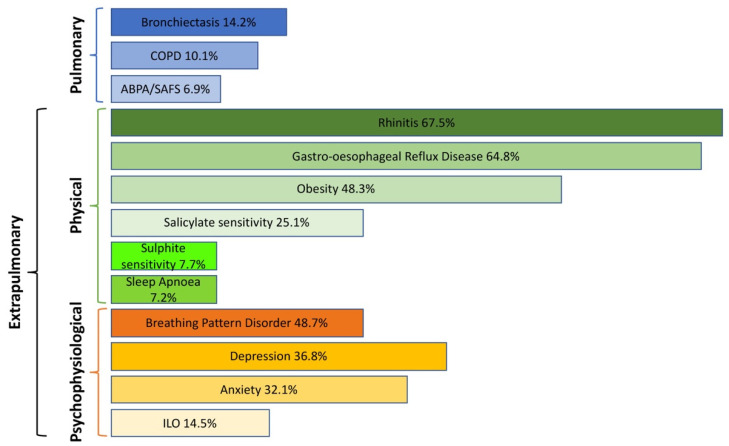
Treatable Traits in the Wessex AsThma CoHort of difficult asthma (WATCH) study. ABPA—Allergic Bronchopulmonary Aspergillosis, SAFS—Severe Asthma with Fungal Sensitisation, COPD—Chronic Obstructive Pulmonary Disease, GORD—Gastro-oesophageal reflux disease, OSAHS—Obstructive Sleep Apnoea-Hypopnoea Syndrome, ILO—Inducible Laryngeal Obstruction.

**Figure 3 jpm-12-01435-f003:**
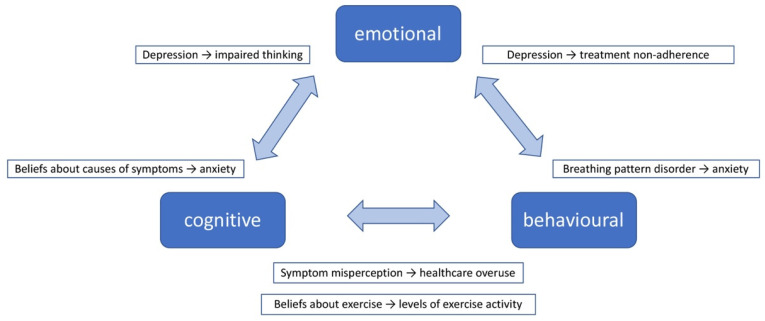
The interaction of emotional, cognitive and behavioural processes with examples in the context of Difficult Breathing Syndrome. Based on Yii and Koch’s framework [40].

**Figure 4 jpm-12-01435-f004:**
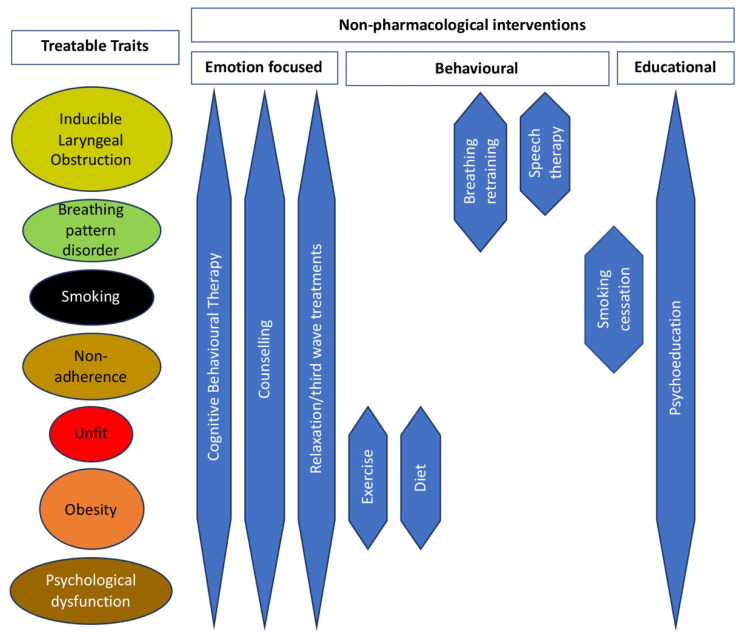
Non-pharmacological treatment approaches for the treatable traits of asthma and “Difficult Breathing Syndrome”.

**Table 1 jpm-12-01435-t001:** Current model of non-pharmacological intervention provision for treatable traits in patients living with Difficult Breathing Syndrome.

*Referral Route & Care Co-Ordination*	*Individual Referral to Each Specialist Decided by Asthma Physician Who also Co-Ordinates Care*								
	Intervention provider								
Intervention level	CBT	Counselling	Relaxation	Exercise	Diet	Breathing training	Speech therapy	Smoking cessation	Psycho- education
Specialist	Clinical Psychologist	Clinical Psychologist/ Asthma Physician	Clinical Psychologist	Physiotherapist	Dietician	Physiotherapist	Speech Therapist	Specialist Behaviour Services-smoking/ Asthma Physician	Clinical Psychologist/ Nurse

**Table 2 jpm-12-01435-t002:** Proposed model of non-pharmacological intervention provision for treatable traits in patients living with Difficult Breathing Syndrome.

*Referral Route & Care Co-Ordination*	*Multidisciplinary Care Plan Drawn up in Collaboration with Patient, Based on Holistic Needs Assessment, Patient and Clinician Preferences. Dedicated Care Co-Ordinator Ensures Agility and Flexibility with Periodic Multidisciplinary Needs Review*								
Intervention provider									
Intervention level (needs based)	CBT	Counselling	Relaxation	Exercise	Diet	Breathing training	Speech therapy	Smoking cessation	Psycho- education
Specialist	Clinical Psychologist	Clinical Psychologist	Clinical Psychologist	Physiotherapist	Dietician	Physiotherapist	Speech Therapist	Specialist Behaviour Services-smoking	Clinical Psychologist/ Nurse
Targeted	CBT therapist/ Psychological Wellbeing Practitioner	Counsellor/ Psychological Wellbeing Practitioner	Counsellor/ Nurse/ Psychological Wellbeing Practitioner	Personal Trainer	Dietician/Specialist Behaviour Services -weight loss	Personal Trainer	Speech Therapist	Specialist Behaviour Services-smoking	Nurse/ Health Coach
Universal	Telehealth/ Self-help	Telehealth/ Self-help/ Asthma Physician	Telehealth/ Social Prescriber/ Health Coach	Self-help/ Social Prescriber/ Health Coach	Self-help/ Social Prescriber/ Health Coach	Telehealth	Speech Therapist	Self-help/ Social Prescriber/ Health Coach/ Asthma Physician	Telehealth/ Social Prescriber/ Health Coach/ Asthma Physician

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
