# Peer review of "Multimorbidity in Difficult Asthma: The Need for Personalised and Non-Pharmacological Approaches to Address a Difficult Breathing Syndrome"

_jpm, 2022, doi:10.3390/jpm12091435_

Round 1
Reviewer 1 Report
Well done, an excellent concept of holistic approach in difficult-to-treat asthma patients. Did you find, after finishing your paper, some limitations of your DBS concept that are still hard to be improved and applied in everydays clinical practice?
Author Response
Response to Reviewer 1:
Reviewer's comment: Well done, an excellent concept of holistic approach in difficult-to-treat asthma patients. Did you find, after finishing your paper, some limitations of your DBS concept that are still hard to be improved and applied in everydays clinical practice?
Response: Thank you for your enthusiasm for our paper and the concept it outlined; we really appreciate the positive feedback. Potential limitations of the DBS concept are an important consideration and your point is very pertinent. In a real-world setting, a key practical limitation to addressing the DBS concept will always be the required multidisciplinary resource and ability to deliver that in an integrated fashion. We have now alluded to this point in the conclusion (lines 828-832).
Reviewer 2 Report
· The introduction is very short and poorly written in its current form. No studies were used to support the aim of this review. What this review adds? Why this review is important in light of previous/recent reviews? Please refines and rephrases the whole introduction using recent evidence.
· Line 41-67; Line 628-657: Strangely, authors refer to case studies. There is no need for that. This is a review paper.
· I understand this is a narrative review, but a method section should be included. It should describe the search terms and types of study designs you used, the databases (e.g., PubMed) you searched, the inclusion/exclusion you addressed....etc.
· Line 147-165; Line 252-267; Line 271-299; Line 374-396: It is important that you include relevant references which support the ideas you are presenting.
· It is unclear whether this review discusses the need to undertake non-pharmacological approaches in both children and adults patients with asthma. The references used may or may not be relevant.
· Many articles related to the topic are missing (J Asthma. 2020 Jan;57(1):105-112; Eur J Gen Pract. 2019 Apr; 25(2): 65-76; Allergy. 2018 Jun; 73(6): 1182-1195; J Allergy Clin Immunol Pract. 2017 Jul-Aug;5(4):928-935; Clin Respir J. 2017 Jan;11(1):13-20; BMC Pulm Med. 2014 Mar 19;14:46; Curr Opin Allergy Clin Immunol. 2020 Feb;20(1):80-84).
· It would be benefit if the authors include other non-pharmacological approaches, such as dietary supplements and/or probiotics, which might affect the risk of asthma or reducing clinical severity. I would suggest authors referring to this article (Encyclopedia 2021, 1(1), 76-86; https://doi.org/10.3390/encyclopedia1010010).
· Ref #82 is very old-please delete.
Author Response
We thank the reviewer for their comprehensive review of our manuscript and their helpful suggestions. We thoroughly considered the reviewer comments and revised our paper accordingly where indicated. As a result we feel that the paper has been significantly enhanced. Please see below for our response to individual reviewer comments:
- Comment 1: The introduction is very short and poorly written in its current form. No studies were used to support the aim of this review. What this review adds? Why this review is important in light of previous/recent reviews? Please refines and rephrasesthe whole introduction using recent evidence.
Response: Thank you very much for your comment on the Introduction that led us to realise that the numbering structure in the original draft was misleading. To clarify, the whole section 1 is the Introduction (lines 40-313 and in this we have used extensive and recent evidence to support the case for this review: we cited 36 papers in the Introduction). We included our rationale for this review in lines 288-313 to outline what this paper adds to the existing knowledge base and why it is important to review the evidence and propose a new model for managing multimorbidity in difficult asthma.
- Comment 2: Line 41-67; Line 628-657: Strangely, authors refer to case studies. There is no need for that.This is a review paper.
Response: Thank you for raising this point. We acknowledge that it may be regarded as unusual to add illustrative cases to a review paper. However, the reason for doing so was to provide clinical relevance and context to the topic under review and illustrate to healthcare professionals how our proposed approach might relate to their patients and clinical practice. We felt this would be particularly appropriate for a journal with the stated aims of the Journal of Personalized Medicine. We therefore respectfully disagree that there is no need for the case to be included.
- Comment 3: I understand this is a narrative review, but a method section should be included. It should describe the search terms and types of study designs you used, the databases (e.g., PubMed) you searched, the inclusion/exclusion you addressed....etc.
Response: Thank you for suggesting including the method for our search as a method section in the paper. We now included the following summary: We performed a narrative review of literature between 2000-2022. Electronic searches MEDLINE, EMBASE, CINAHL, AMED, PsychINFO, Cochrane Airways Group Centralised Register, EMCARE, PubMed, PsychARTICLES, Francis and Taylor online, Elsevier, ScienceDirect, Sage, Google Scholar, EBSCOhost. Additionally, hand-searches of Journal of Multimorbidity and Comorbidity were performed. Search terms for primary disease included asthma, chronic obstructive pulmonary, chronic lung disease, respiratory, disease, respiratory disease, respiratory illness. Secondarily, we included searches for the extrapulmonary treatable traits of asthma, such as vocal cord dysfunction, anxiety, depression, psychological comorbidities, psychological distress, psychological dysfunction, breathing pattern disorder, inducible laryngeal obstruction, obesity, gastro-oesophageal reflux disease or GORD and (non-)adherence. We searched specifically for non-pharmacological interventions in extrapulmonary treatable traits in asthma, including terms: non-pharmacological, exercise, dietary, pulmonary rehabilitation, surg*(ical), smoking cessation, psychoeducation. We excluded papers with non-adult population and papers that included chemical agents as we considered these as other pharmacological interventions. All reviewed papers were published in English.
- Comment 4: Line 147-165; Line 252-267; Line 271-299; Line 374-396: It is important that you include relevant references which support the ideas you are presenting.
Response: Thank you for your pertinent comment which we fully agree with. We can confirm that we included supportive references to the ideas we presented. We are sorry that this was not immediately evident and have amended the text to highlight this more clearly. Specific responses about the sections of text highlighted are given below:
- Comment 4a: Line 147-165;
Response: We can confirm that the description outlined in this section refers to the material contained in the landmark paper cited at the beginning of the section (reference 18). We have further clarified this point by additional referencing to that paper in this section.
- Comment 4b: Line 252-267;
Response: This section expanded on the two papers cited. We re-read the section and made the following modification: New text: “Smoking status, Inhaler technique, Monitoring, Pharmacotherapy, Lifestyle, Education and Support (SIMPLES) assessment was introduced as a tool for use in primary care to support management of patients with poorly controlled asthma [33]. It encompassed self-management, education, monitoring and lifestyle changes (with emphasis on smoking status) in addition to pharmacotherapy. Often ignored facets such as regular review and accessibility were also recognised and given prominence. This was coupled to guidance on when to refer from primary to specialist care. Another important component to SIMPLES was the early adoption of digital technologies with web-based access to both the SIMPLES framework and relevant assessment tools”.
We believe that there is no need to add further references for describing the merits of the Severe Asthma Toolkit. It should now be apparent that the cited paper supports the text description.
Comment 4c: Line 271-299;
Response: We thank the reviewer for identifying this gap in supporting referencing and have now added 4 references to support the statements:
- Specialised respiratory services (adult) – Severe asthma. https://www.england.nhs.uk/publication/specialised-respiratory-services-adult-severe-asthma/ – reference 30.
- Hew M, Menzies-Gow A, Hull JH, Fleming L, Porsbjerg C, Brinke AT, Allen D, Gore R, Tay TR. Systematic Assessment of Difficult-to-Treat Asthma: Principles and Perspectives. J Allergy Clin Immunol Pract. 2020 Jul-Aug;8(7):2222-2233. doi: 10.1016/j.jaip.2020.02.036. Epub 2020 Mar 12. -reference 35
- McDonald VM, Vertigan AE, Gibson PG. How to set up a severe asthma service. Respirology. 2011 Aug;16(6):900-11. doi: 10.1111/j.1440-1843.2011.02012.x. - reference 36
- McDonald VM, Harrington J, Clark VL, Gibson PG. Multidisciplinary care in chronic airway diseases: the Newcastle model. ERJ Open Res. 2022 Aug 15;8(3):00215-2022. doi: 10.1183/23120541.00215-2022. -reference 37
- Comment 4d: Line 374-396:
Response: In these sections we endeavoured to provide our readers with some overarching key papers that cover many of the approaches described here. The purpose of doing this was to help the reader familiarise with relevant concepts from a few key integrated sources instead of multiple sources each focusing on one of the listed approaches.
- Comment 5: It is unclear whether this review discusses the need to undertake non-pharmacological approaches in both children and adults patients with asthma. The references used may or may not be relevant.
Response: Thank you for pointing out that it remained unclear that this review focused only on holistic non-pharmacological approaches to address multimorbidity for adults with difficult-to-treat asthma. We have now clarified this in the paper (lines 70 and 76) and hopefully the method section also clarifies this for our readers (line 330).
- Comment 6: Many articles related to the topic are missing
Response: Thank you for highlighting further potentially relevant papers to include in this review. It should be noted that the focus of our review paper was on non-pharmacological approaches to multimorbidity within the Difficult Breathing Syndrome of difficult-to-treat asthma and that most of these papers do not specifically address that scenario. However, where the suggested papers were deemed to have applicable relevance we have included them as detailed below.
(J Asthma. 2020 Jan;57(1):105-112; This is now cited (line 538 ).
Eur J Gen Pract. 2019 Apr; 25(2): 65-76; This is now cited (line 538).
Allergy. 2018 Jun; 73(6): 1182-1195; This is now cited (line 797).
J Allergy Clin Immunol Pract. 2017 Jul-Aug;5(4):928-935; This is now cited (line 595).
Clin Respir J. 2017 Jan;11(1):13-20; Thank you for suggesting this paper. However we do not think that this is a technique that is of direct relevance to addressing multimorbidity in difficult-to-treat asthma, which is the focus of this review. It is also not a widely applied treatment technique. This citation therefore has not been included.
BMC Pulm Med. 2014 Mar 19;14:46; Thank you for suggesting this paper however we think discussing the particular challenges of non-pharmacological management of Difficult Breathing Syndrome in pregnancy warrants a standalone paper and we did not feel it gives credit to this important topic just to mention it in this paper. Therefore we decided to exclude this topic.
Curr Opin Allergy Clin Immunol. 2020 Feb;20(1):80-84). This is now cited (line 596).
- Comment 7: It would be benefit if the authors include other non-pharmacological approaches, such as dietary supplements and/or probiotics, which might affect the risk of asthma or reducing clinical severity. I would suggest authors referring to this article (Encyclopedia 2021, 1(1), 76-86; https://doi.org/10.3390/encyclopedia1010010).
Response: Thank you for your suggestion to include this paper. Again we refer to the fact that the purpose of this paper was to review non-pharmacological approaches to address multimorbidity in difficult-to-treat asthma. We made a conscious decision to exclude the interventions with such supplementary substances as we considered them pharmacological in nature. This is now made clear in the paper (lines 330-332).
- Comment 8: Ref #82 is very old-please delete.
Response: Thank you for suggesting this change. We deleted the old reference that was a key paper and added a more recent paper by the same author that still described the original theory that is fundamental in our proposed approach. This is listed in the amended paper as reference 85.
Round 2
Reviewer 2 Report
No further comments.